# Flood simulation with the RiverCure approach: The open dataset of the Águeda 2016 flood event

Ana M. Ricardo[1], Rui M.L. Ferreira[3], Alberto Rodrigues da Silva[2], Jacinto Estima[2, 4], Jorge Marques[2], Ivo Gamito[2], Alexandre Serra[2]

[1]CERIS - Civil Engineering Research and Innovation for Sustainability, 1049-001 Lisbon, Portugal
[2]INESC-ID, Instituto Superior Técnico, Universidade de Lisboa, 1049-001 Lisbon, Portugal
[3]CERIS, Instituto Superior Técnico, Universidade de Lisboa, 1049-001 Lisbon, Portugal
[4]CISUC, Departamento de Engenharia Informática, Universidade de Coimbra, 3030-790 Coimbra, Portugal

*Correspondence to*: Ana M. Ricardo (ana.ricardo@tecnico.ulisboa.pt)

**Abstract.** Floods are among the most common natural disasters responsible for severe damages and human losses. Numerically produced data, managed by user-friendly tools for geographically referenced data, has been adopted to increase preparedness and reduce vulnerabilities. This paper describes the locally sensed and numerically produced data that characterize a flood event occurred in February 2016 in the Portuguese Águeda river, shortly referred as Agueda.2016Flood. The data was managed through the RiverCure Portal, a collaborative web platform connected to a validated shallow-water model featuring modelled dynamic bed geometries and sediment transport. The dataset provides a synthesis of topo-bathymetric, hydrometric and numerically-produced data from a calibrated hydrodynamic model. Due to the lack of measured hydrometric data near the city, the numerically produced data is crucial for the complete description of the flood event. The Agueda.2016Flood dataset constitutes a complete validation test for flood forecasting models and a tool to better mitigate floods in this river and in similar rivers. Thus, Agueda.2016Flood is a relevant dataset for River Águeda stakeholders as well as for the community of flood modellers, as it provides a well-documented validation event for forecasting tools.

## 1 Introduction

Floods cause widespread damages and human losses. For instance, according to the Centre for Research on the Epidemiology of Disasters, floods were the most frequent type of natural disasters between 1998 and 2017 (Wallemacq et al., 2018). There have been changes in the frequency and magnitude of floods (Amponsah et al., 2018), including an increase in their severity in central and north Europe (Blöschl et al., 2019; Hall et al., 2015). Recently, severe floods were observed in Germany and Belgium in the summer of 2021 (Fekete and Sandholz, 2021). Accurate forecasting tools and early warning systems have been proposed to increase preparedness and reduce systemic vulnerabilities (Yuan et al., 2022).

While planning for flood resilience differs from nowcasting (Yuan et al., 2022), both activities rely on analysing available hydrological data and hydraulic modelling, even at different time scales. Crucially, both require decision support systems that channel available input data, conduct hydraulic modelling, possibly combined with the assimilation of input data, and collect

and organize results into datasets ready to be interpreted by decision-makers. Fostering the Digital Earth concept, those benefit from the current research efforts in designing and developing digital earth twin hydrology systems (DARTHs), which are not isolated models but services or components that can be assembled together (Rigon et al., 2022).

This paper describes a synthesis of data that allows the spatial and temporal characterization of a flood occurred in the Portuguese Águeda river in February 2016. It also describes the main features of the RiverCure Portal (RCP), the information system employed to manage and articulate the locally sensed data and the numerically produced data. The RiverCure Portal (http://rivercure.inesc-id.pt/) is a web platform aligned with the DARTHs paradigm and developed to integrate hydrodynamic and morphodynamic modelling tools and input data (Rodrigues da Silva et al., 2023).

The Águeda river runs through the city of Águeda, that has registered many flood occurrences causing significant damages. As a prone to flood area, Águeda municipality had made considerable investments in flood protection. Namely, in 2015, it invested around two million euros in constructing a secondary river channel to divert the river flow. However, those efforts were not enough to prevent a severe flood event in February 2016, which was the most significant flood of the previous 15 years (https://www.jn.pt/local/noticias/aveiro/agueda/agueda-com-maiores-cheias-dos-ultimos-anos-5027652.html). That flood revealed weaknesses in the flood defence infrastructure of the municipality although the water level in the river, in the city vicinity, was, at its peak, from 10 to 20 cm below the protection wall crest. That flood event was the result of heavy precipitation associated to strong instabilities in the North Atlantic, causing significant disturbances in the city of Águeda, affecting its mobility, public services, and infrastructures.

Although flooding is a recurrent problem of Águeda city, there are no curated datasets that allow for the description of its flooding events. This paper aims at fulfilling that gap assembling the key input geographical and hydrologic data, the pre-processing steps and the rationale for establishing the parameters of the hydrodynamic modelling tools that ultimately produce the hydrometric datasets of the aforementioned February 2016 flood event. To have a complete flood description, the numerically produced hydrometric data (flow depths, discharges and flow velocities) are required due to the lack of reliable physical data as the hydrometric sensor installed at Águeda bridge was malfunctioning during the flood event. Thus, the presented dataset is a unique synthesis of topo-bathymetric, hydrological and numerically produced hydrometric data. It is very relevant locally as it documents an event relevant for River Águeda stakeholders. It is also relevant for a wider community of flood modellers as it provides a well-documented validation event for flood models, particularly, shallow-water models. The community can also use this dataset as a starting point to design independent experiments and tools, and as a training example for more general usage of the RiverCure Portal .

## 2 Supported software tools

The software tools used in the scope of this research are the RiverCure Portal and the HiSTAV numerical model, which run in an integrated way.

The RiverCure Portal (RCP) is a collaborative web application, publicly available at http://rivercure.inesc-id.pt/, that allows different organizations to setup and manage their geographic contexts, sensors and events in a controlled way. RCP has been developed with Web and GIS technologies, such as Django, GeoDjango, PostgreSQL and PostGIS. RCP conveys geographical

and hydrological data to a numerical model, as a processing service to provide simulation results. A detailed description of the concepts, design and technologies behind RCP is presented by Rodrigues da Silva et al. (2023). Herein, only a brief description of the main features are provided.

The structure of RCP consists of two main top-level concepts: Contexts and Sensors. The Context allows defining the geometry of the area and managing the events to be modelled and simulated. The geometrical details consist of the set of georeferenced

features that are visualized and edited over a basemap from OpenStreetMaps (Haklay and Weber, 2008) through the Leaflet (https://leafletjs.com/) library (Peterson, 2014 and Edler and Vetter, 2019). The polygons and polylines can be either uploaded (i.e., when they are developed offline) or directly defined in the RCP. The raster files containing the digital elevation model (DEM) and the roughness coefficients should be uploaded. The geographic features of the Context include the mesh generation process, as the mesh is a required feature of the numerical modelling. The mesh is generated from the geometrical and raster

input data and employs the open-source 3-D finite element grid generator Gmsh (Geuzaine and Remacle, 2009). The connection between the RCP and the numerical model is established by defining an event to be modelled. The Event features allow the user to define the results writing rates and the simulation's starting and ending time.

The Sensors represent cyber-physical devices that load and store the environmental data in the RCP. Generically, a Sensor reads and collects data in a given location, usually a geographical point, with a given frequency. The Sensor Observations

(data values) are automatically obtained or manually introduced in the RCP, by users, either on a value-by-value basis or in sets of values by a given spreadsheet. The mapping between a Sensor and a Context is defined by associating a Sensor with a boundary point or an entire boundary of a Context.

The numerical model currently linked to RCP is HiSTAV (High Performance Computing version of the original Strong Transients in Alluvial Valleys model) however it could use other existing simulation tools. HiSTAV is an in-house shallow-

water model featuring dynamic bed geometries and sediment transport modelled via the difference between capacity bedload discharge and local solid discharge (Ferreira et al., 2009; Canelas et al., 2013; Conde et al. 2020, Conde, D. 2018). The model comprises the total mass conservation equation, the momentum conservation equation and the sediment mass conservation equation. The conservation equations system is hyperbolic and discretized with a finite-volume method employing a flux-splitting technique and a reviewed Roe–Riemann solver with appropriate source-term formulations to ensure full

conservativeness (Conde et al., 2020). The sediment mass conservation expresses the interaction between the bed and the flow in the bed.

In HiSTAV, the closed conservation equations are discretized by a Finite Volume approach and solved explicitly, obeying a Courant-Friedrichs-Lewy (CFL) condition, a conventional condition to restrict the computational time step. The implementation is entirely cross-compatible between CPUs and GPUs, through an intuitive object-oriented approach (Conde

et al., 2020). It thus supports distributed and heterogeneous computing of significant problems at very high resolutions.

Therefore, there are no special requirements for the machines that may run HiSTAV; it can spread its load through available GPUs or CPUs. The current simulations were run on a multi-CPU/GPU computational server with two nodes, each with two Xeon(R) E5-2650 Octa-core CPUs and 4 NVIDIA GPUs, with 128GB ECC RAM installed and a 1TB de 10Krpm hard disk TYAN FT77A-B7059 2013.

The data produced by HiSTAV is stored in *.vtk* files created by the Visualization Toolkit (VTK), state-of-the-art open-source software for manipulating and displaying scientific data (https://vtk.org/).

## 3 Data records

### 3.1 Overview of the dataset

The simulation of Águeda 2016 flood event involved a geographical extent corresponding to an area of 560 ha crossed by a

9.8 km long stretch of the Águeda river, including the riverfront part of Águeda city, as represented in Figure 1. Regarding the time span, this flood event was defined between 09/02/2016 00:00:00 and 16/02/2016 23:00:00.

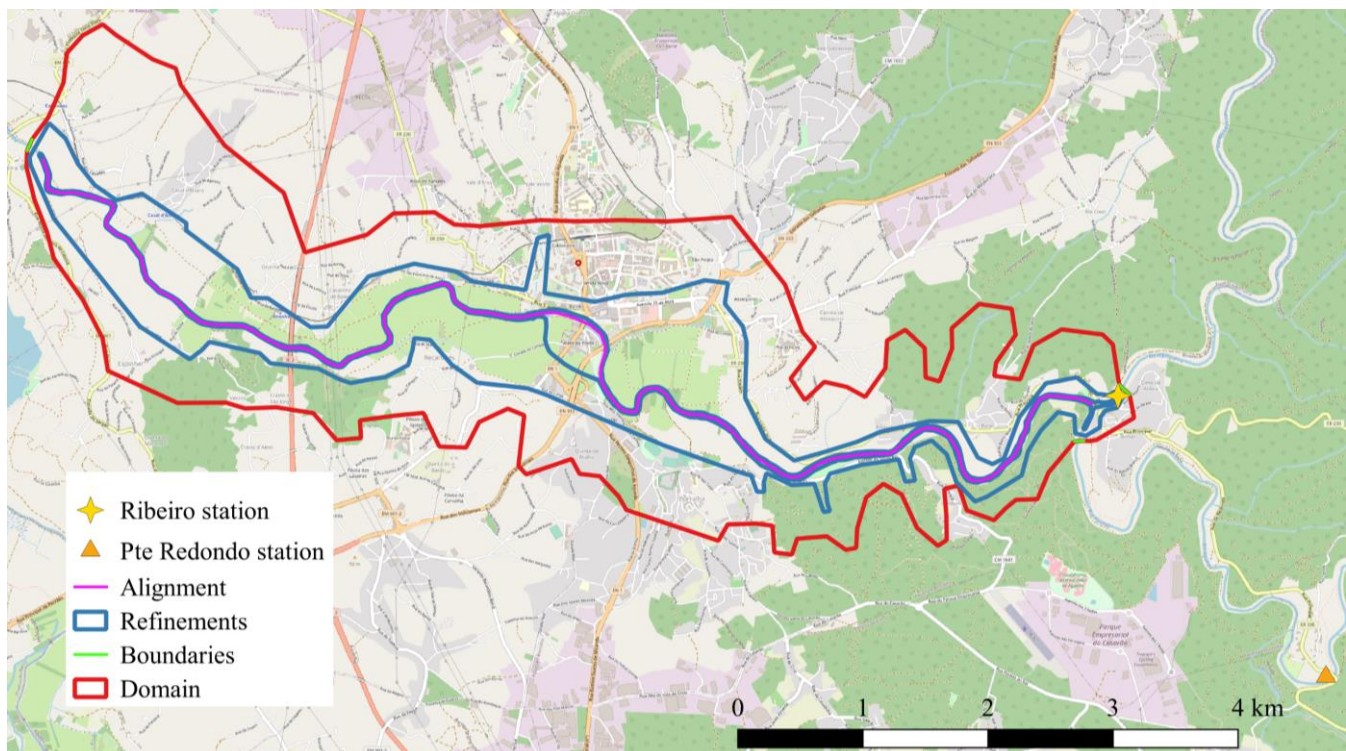

**Figure 1 Map of the modelled area (domain) including the geometrical features for mesh generation (alignment and refinements) and the location of the inlet and outlet boundaries. The orange triangles point the location of the hydrometric stations (Ribeiro and**
**Ponte Redonda). The map was produced with QGIS employing the basemap from ©OpenStreetMap.**

Figure 2 presents an overview of Agueda.2016Flood dataset. The dataset includes the geometrical features of the simulated context and two main subsets of data: the field data, including topo-bathymetric, land-use and hydrologic data collected to

feed the flood event simulation, and the numerically produced hydrometric data. Details on each part of the dataset are provided in the following subsections.

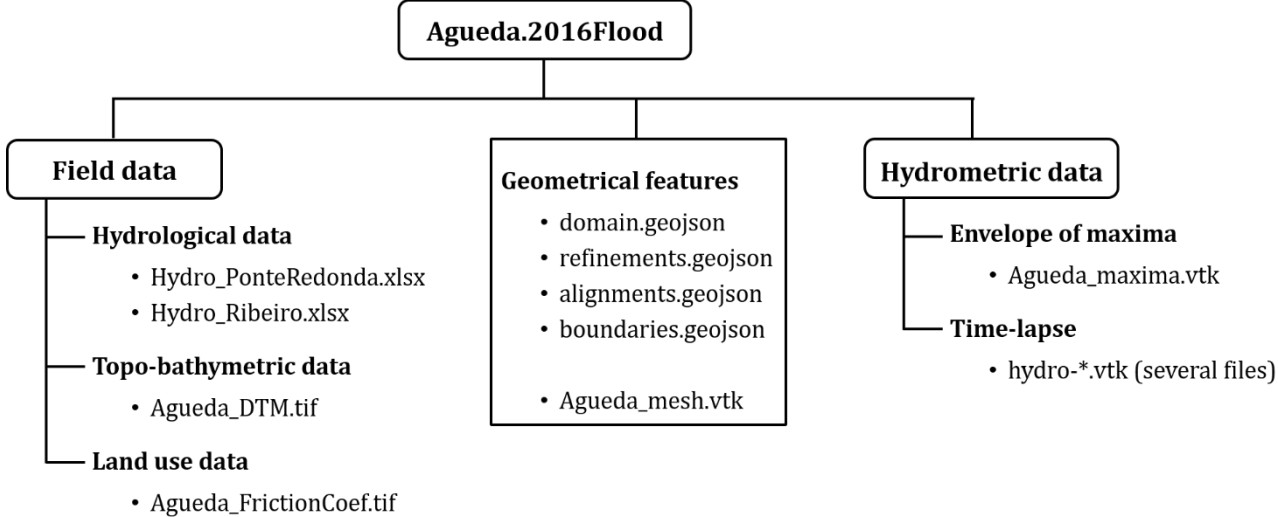


**Figure 2 - Schematic representation of the dataset organization with the identification of the files included in each part.**

The dataset Agueda.2016Flood (Ricardo et al., 2022) is publicly available on the repository HydroShare. The specification of the employed data entities uses the controlled natural language RSL, as discussed in Rodrigues da Silva and Savić (2021) (see Appendix A for details on the main concepts and properties related to the Context definition).

**3.2. Geometrical features**

The subset named as *Geometrical features* contains auxiliary elements rather than actual data. These elements include the features to geometrically define the simulation and its mesh.

Concerning the geometrical characteristics of the simulation, the numerical model requires a set of lines and polygons in GeoJSON format, namely:

- domain: a polygon delimitating the area to be modelled;
- refinements: polygons delimitating areas where specific mesh refinements are desired;
- alignments: a polyline identifying the river centreline;
- boundaries: polylines aligned with domain limits defining the inlet and outlet boundaries of the domain.

The boundaries are characterized by the type of boundary condition (input, known output, critical or transmissive) and when
applicable the type of data to be given as input to the model (e.g., depth, discharge, velocity or elevation). These features can be set or updated through the user interface of RCP. The polyline defining the main channel alignment and the refinement polygons are not mandatory, they are used for custom made improvements on the mesh size definition. A feature named 'CL', storing an indicative value for the length of the largest cell size in meters must be an attribute of the polygons defining the

domain and the desired mesh refinement zones as well as of the polyline defining the main channel alignment. Furthermore,

to fulfil the mesh generator requirements, the geometries cannot intersect, i.e., any larger polygon must completely contain a given polygon or polyline.

The geometrical features subset also includes the file *Agueda_mesh.vtk* containing the mesh used in the simulation of the flood event. The mesh consists of a set of unstructured triangular cells containing the following data: cell ID (named Address, [-]), bed level ([m]), roughness coefficient ($[m^{1/3}s^{-1}]$), and physical tag to identify the cells corresponding to boundary cells ([-])

and the length of the largest cell size ([m]).

### 3.3 Field data

The field data subset corresponds, in a broad sense, to data locally collected. It includes local topography, river bathymetry, land-use data and hydrologic data obtained from two hydrometric stations.

### 3.3.1. Hydrological data

The hydrological data employed as model input data consist in hourly discharge time-series obtained from two hydrometric stations managed by the Portuguese Environmental Agency (APA, https://snirh.apambiente.pt/), Ribeiro and Ponte Redonda stations (Table 1). Ribeiro station is located within the simulated area in the Alfusqueiro river, close to the its confluence with Águeda river. Ponte Redonda station is located in Águeda river, 5 km upstream of the domain boundary. Both stations are equipped with automatic sensors that measure the hydrometric level at every hour, except when a critical threshold is reached.

In that case, the measurement frequency is increased (Pereira, 2021). The measured flow levels (*h*) are converted to discharges (*Q*) by the rating curves defined and updated by the national authorities. Further details on the hydrometric stations and the corresponding rating curves can be obtained in APA website or in Pereira (2021).

A synthetic and constant time series, with 20 hours of length and discharge equal to the value obtained for 09/02/2016 00:00, was added to the event data series to ensure the model warm up. Table 1 summarizes the information regarding the two

hydrometric stations.

**Table 1 – Information on the name of the file containing the discharge data, the location and the rating curves of each station.**

| Name | Data file | Location (ETRS89 / Portugal TM06 coord. system) | | Rating curve |
| --- | --- | --- | --- | --- |
| | | **Latitude** | **Longitude** | |
| Ribeiro | Hydro_Ribeiro.xlsx | 40.566 | -8.398 | $Q = 16.222 \times (h - 0.805)^{2.39}$ for $0.80 \text{ m} < h < 1.83 \text{ m}$ |
| | | | | $Q = 4.632 \times h^{2.168}$ for $1.83 \text{ m} < h < 4.5 \text{ m}$ |
| Ponte Redonda | Hydro_PonteRedonda.xlsx | 40.546 | -8.378 | $Q = 11.079 \times (h - 0.877)^{2.35}$ for $0.877 \text{ m} < h < 2.22 \text{ m}$ |
| | | | | $Q = 1.124 \times (h + 0.64)^{2.84}$ for $2.22 \text{ m} < h < 5 \text{ m}$ |

Another hydrometric station exists within the simulated area, Ponte de Águeda station, located in the bridge that crosses the river in Águeda downtown. The data from that station revealed to be inaccurate due to any eventual malfunctioning of the sensor as well as due to the lack of update of the rating curves after the regulation works carried out in 2015 in the river banks on the vicinity of Águeda city.

### 3.3.2. Topo-bathymetric data

The ASTER Global Digital Elevation Model (GDEM) Version 3 (ASTGTM), publicly provided by NASA (https://search.earthdata.nasa.gov/) was the base for the topographic characterization of the modelled area. The ASTER platform has three subsystems: a visible and near-infrared radiometer, a shortwave-infrared radiometer, and a thermal-infrared radiometer (Yamaguchi et al, 1998; Fujisada et al., 2018). Version 3 counts with a technical methodology for improving the initial tile-based waterbody data (Fujisada et al., 2018). The geographic coverage of the ASTER GDEM extends from 83° North to 83° South, divided into 22912 tiles with a spatial resolution of 1 arc second (approximately 30-meter horizontal posting at the equator). Each tile is distributed in GeoTIFF format and projected on the 1984 World Geodetic System (WGS84)/1996 Earth Gravitational Model (EGM96) geoid. The tile ASTGTMV003_N40W009, which includes the littoral centre and North of Portugal, was downloaded, and the interest region was cropped and employed as a starting point for the digital elevation model (DEM) provided by the raster file *Agueda_DEM.tif*. The base raster was edited for accuracy and resolution improvement by adding detail on the river bottom and banks, particularly on the vicinity of Águeda city.

The river centreline and width were defined through Google Earth. River Águeda in front of Águeda town has suffered many interventions that resulted in a channel with a rectangular cross-section with protection walls. A significant effort was carried out to identify adequate and accurate data from elements provided by Águeda municipality and to collect field data regarding the elevation of the protective walls of the riverfront of Águeda city and regarding the river bathymetry. Since the city was flooded not because of the protection walls overtopping but because of vulnerabilities in these walls, these had to be investigated to be incorporated in the DEM reproducing the geometry of the under-passages of Águeda bridge.

Although based on the relatively poor resolution of the global ASTER-DEM, the produced DEM ensured the accuracy and the adequacy of the spatial resolution in what concerns the river bottom and the city on the river vicinity.

### 3.3.3. Land-use data

The spatial distribution of the roughness coefficient over the modelled area is also a required model input. It was obtained from the COS2018, a spatial dataset produced by the Portuguese Governmental office for territorial development, DGT (Direção Geral do Território - https://www.dgterritorio.gov.pt/dados-abertos), representing the thematic map of land use and land cover for mainland Portugal for the year 2018. COS2018, made available as Linked Open Data (LOD) in RDF format, is a map of polygons with a defined minimum cartographic unit (1 ha) with a distance between lines equal to or greater than 20 m. It is based on photo interpretation and has a terminology with more than 80 classes. Manning's coefficient values (*n*) used for specific land cover are based on the well-established values presented in Chow (1959) and van der Sande et al. (2003). The

raster file *Agueda_FrictionCoef.tif* includes the spatial distribution of the Manning-Strickler friction coefficient, $K_s = \frac{1}{n}$ [m$^{1/3}$s$^{-1}$].

## 3.4 Hydrometric data

The hydrometric data to characterize the flood event was numerical produced by HiSTAV. This subset of the dataset includes the spatial distribution of the maximum values of the hydrometric variables and instantaneous maps for a selected set of time

instants.

### 3.4.1 Envelope of maxima

The file *Agueda_maxima.vtk* corresponds to the envelope of maxima for each variable, i.e., stores the maximum values of each mesh cell for the modelled flow variables. The variables included are: flow depth, water level, flow velocity, hazard index (defined as $h \times (v + 0.5)$ where $h$ is the flow depth, and $v$ is the flow velocity according to Penning-Rowsell et al., 2005), the

time to reach the maxima hazard index of each cell and the time to reach the wet state of each cell.

The rate of update of maxima values during a given simulation can be set by the user on the RCP. In the present simulations, the maxima were updated every second.

### 3.4.2 Time-lapse

The spatial distribution of bed level ([m]), water level([m]) and flow velocity ([m/s]) for a set of time instants are provided by

the *hydro-\*.vtk* files. The numbers in the file name indicate the simulation time step.

In the present case, the time dependent result files were written hourly but sampled every three hours as a compromise between a detailed time characterization and storage management. The hourly dataset will be shared upon request directly to the authors.

## 4 Data validation

The accuracy of the discharge time series computed from the data collected by the Ribeiro and Ponte Redonda hydrometric

stations was confirmed by Pereira (2021) by hydrological and hydraulic modelling employing rain data from radar and from rain gauges.

The fixed bed version of HiSTAV requires the empirical quantification of one parameter, to complete the roughness closure. In this case, it is the conveyance coefficient (the inverse of a roughness coefficient) of the Manning-Strickler equation. The model was given maps of the Manning-Strickler that were merged with the mesh at the pre-processing stage. No formal

calibration was necessary. A bulk verification of the roughness parameters and topo-bathymetry was undertaken: it was verified that the roughness and topo-bathymetric data led to the observed river water level in the vicinity of Água at t = 110 h (12/02/2016 18:00:00), i.e.10-20 cm below the protection wall crest.

The validation tests grant the quality of the numerically produced data shown by the HiSTAV tool. The HiSTAV model has been compared with theoretical solutions and experimental data for both resistance and solid transport (Canelas et al., 2013)

and was tested in a benchmark text of 2D dam-break flows over a sand bed (Soares-Frazão et al., 2012). HiSTAV has also been employed to model a tsunami in the Tagus estuary (Lisbon), demonstrating a good performance on the Monai Valley benchmark and in comparison with historical data (Conde et al., 2015).

The Águeda 2016 flood event was largely reported on the Portuguese media allowing to gather informal but accurate data regarding water depth and extent of flooded areas. The efforts on improving the topo-bathymetric data allowed to obtain a

good agreement between the modelled and observed flood extent in Águeda downtown.

The 2016 flood event is of particular interest due to the large media coverage allowing to gather informal but accurate data regarding water depth and extent of flooded areas. In this event, the water depth in the river, near the bridge was, at its peak, from 10 to 20 cm below the protection wall crest. This has been observed by local authorities and registered in photos and videos between 4 and 8 PM on the 12th February. These photos were used to verify the extent of inundation along Rua Luís

de Camões and Rua Vasco da Gama. Examples of the photos collected and analysed to compared with the numerically produced data are shown in Figs. 3 and 4.

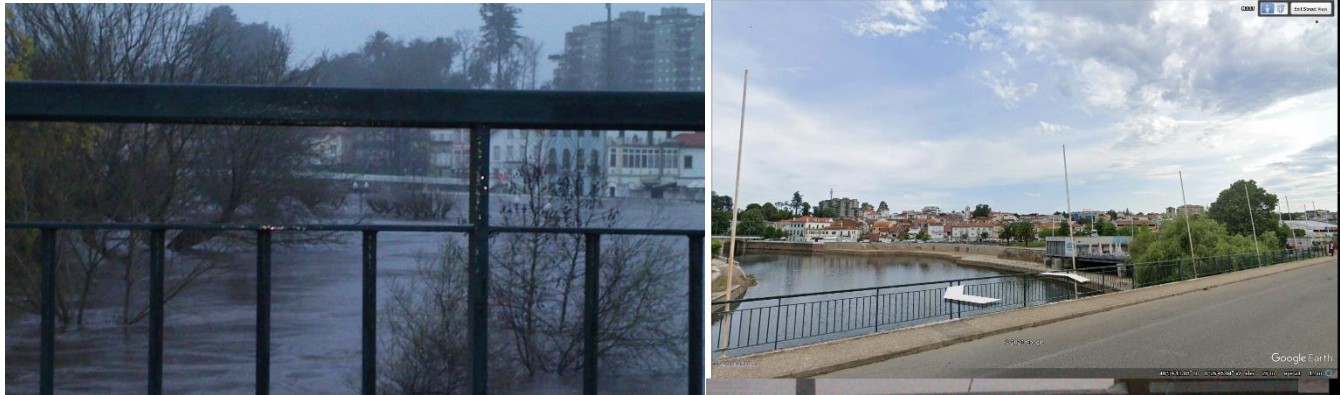

**Figure 3 Photo of the water level in the river (left) and Google Earth Pro view (street view) from the location where the photo was taken.**

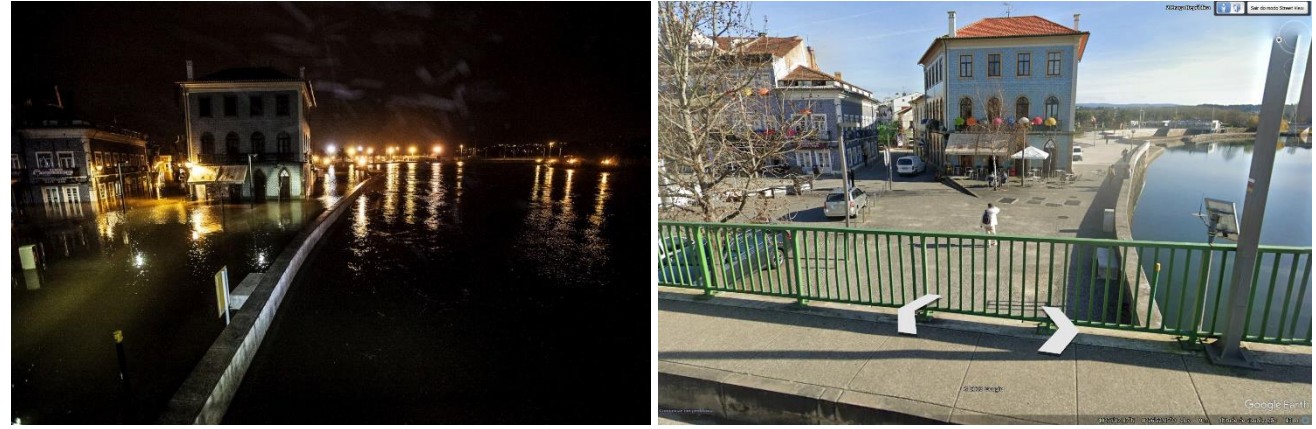


**Figure 4 Photo of the flood in a square (Praça da Repúlica) and a street (Rua Vasco da Gama) in Águeda downtown and Google Earth Pro view (street view) from the location where the photo was taken.**

Figure 5 presents a 2D distribution of the maximum water depth values on the modelled domain overlapped on the terrain

model.

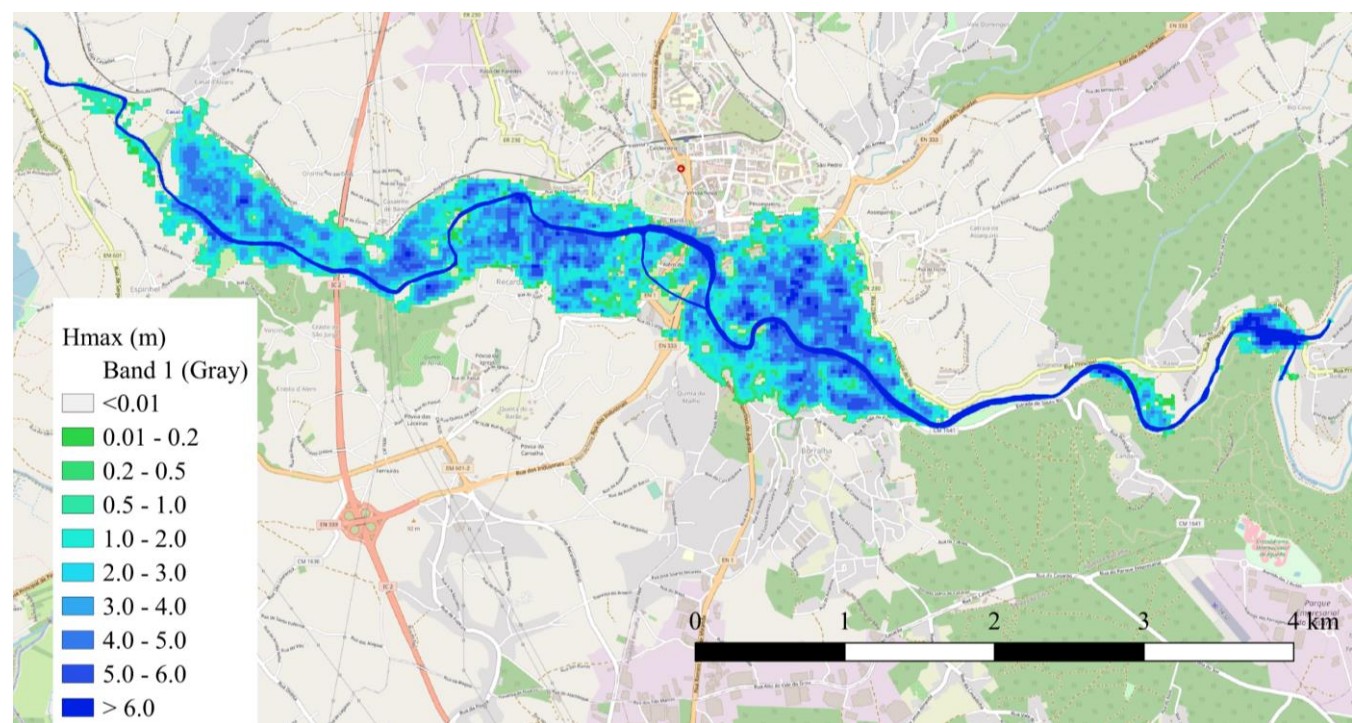

**Figure 5 Distribution of the maximum values of the computed water depth overlapped on the map. The colour scale is in meters (m). The map was produced with QGIS employing the basemap from ©OpenStreetMap.**

Another way to verify the reliability of the produced dataset is to compare the input discharge data with the numerically produced data, as shown in Fig. 6. The "upstream" hydrograph corresponds to the numerically produced discharge on the cross section at approximately 400 m downstream of the domain limit at the intersection of the two inlet channels. The discharge computed in that section coincides with the summation of the two input hydrographs. The "downstream" hydrograph was obtained at the domain outlet boundary. The comparison of the two computed hydrographs shows the expected delay and
damping of the flood from the entrance to the outlet of the domain.

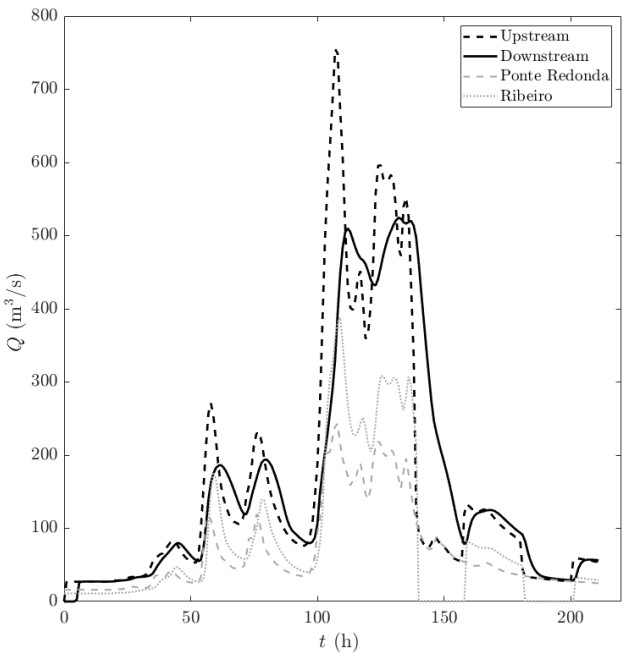

**Figure 6 Tine series of flow discharge at the inlet boundaries (Ponte Redonda and Ribeiro), at the confluence of the two inlet streams (Upstream) and downstream boundary. The 'Upstream' and 'Downstream' hydrographs were computed from the numerically produced database. The time instant *t=0* h corresponds to 04:00 am of the 8th February 2016.**

## 255   5 Data and code availability

All        data        is        store        on        the        Agueda.2016Flood        dataset (http://www.hydroshare.org/resource/937927473a3a4e66a07a2e2fdd9d581e, Ricardo et al., 2022), that is publicly available on HydroShare repository. HydroShare is an open source web-based system developed, mainly, for water related professionals to easily share, collaborate and publish all types of scientific data. HydroShare adheres to the FAIR archival standard and
provides a Representational State Transfer (REST) Application Program Interface (API) that allows third-party applications to interact with HydroShare resources. Hydroshare is a large Python/Django application with some extra features and technologies added on. All data uploaded to HydroShare become part of a "resource" which provides the ability to group

together multiple files of different types in one location. The dataset presented herein is the resource named Agueda.2016Flood. The resource contents is organized in three folders following the schematic representation of Figure 2. The download of the files is available by scrolling down on the resource's landing page until the Content section. It can be done by downloading each file individually, by downloading a file zipped or by download all content as zipped bagit archive. Further details are provided at https://help.hydroshare.org/creating-and-managing-resources/view-and-download-a-resource/.

The RiverCure Portal is publicly available at http://rivercure.inesc-id.pt/ and can be used to run the presented case study or any other case upon a free registration. The numerical model HiSTAV can be explored through the RiverCure Portal. Conde et al. (2020) provides further details about the numerical model, its discretization and its implementation within the paradigm of High Performance Computing (HPC) for GPU and CPU-based computer architectures.

## 6 Conclusions

This work describes the dataset Agueda.2016Flood. It presents a unique and highly relevant dataset to fully characterize the flood event occurred in February 2016 in the Portuguese Águeda river, a river for which there are no curated datasets of flooding events, despite its relatively high frequency. The dataset was managed through the RiverCure Portal, a collaborative web platform connected to HiSTAV, a validated shallow-water model featuring modelled dynamic bed geometries and sediment transport.

The dataset includes hydrological, topo-bathymetric and land use data, and numerically-produced hydrometric data from a calibrated model. Per se, none of those data subsets would be sufficient, it is the synthesis that allows the complete description of the flood event. Thus, the dataset constitutes a relevant and complete validation test for other flood forecasting models and a tool to better understand and mitigate floods in this river and in similar rivers.

## Appendix A

Below it is illustrated the data model supported by the RiverCure Portal that shows its main concepts and properties related the Context definition. This specification is defined according to the ITLingo RSL language (Rodrigues da Silva and Savić, 2021).

```
// Contexts
DataEntity e_Context "Context": Master [
    attribute id "Id": Integer [constraints (PrimaryKey)]
    attribute code "Code": String [constraints (NotNull Unique)]
    attribute Name "Name": String [constraints (NotNull Unique)]
    attribute organization "Organization": Integer [constraints (ForeignKey(e_Organization))]
    attribute creator "Creator": Integer [constraints (ForeignKey(e_User))]
    attribute createDate "Create Date": Datetime
    attribute isPublic "Is public" : Boolean [defaultValue "False"]
    attribute hydrofeature "Hydro Feature": Integer [constraints(NotNull ForeignKey(e_HydroFeature))]
    attribute geomExternalBoundary "Geometry External Boundary": GeoPolyline  // domain ?
    attribute CLExternalBoundary "CL External Boundary": Decimal
    attribute hasMesh "Has mesh" : Boolean [defaultValue "False"]
```

```
            attribute taskId "Task id": String
attribute requester "Requester": Integer [constraints (ForeignKey(e_User))] ]
        // Context's Geo Features
        DataEntity e_ContextDTM "Context DTM": Parameter [
            attribute id "Id": Integer [constraints (PrimaryKey)]
            attribute context "Context": Integer [constraints(NotNull ForeignKey(e_Context))]
attribute rasterFile "Raster File": FilePath
            attribute raster "Raster": GeoRaster   ]

        DataEntity e_ContextFrictionCoeff "Context Friction Coefficient": Parameter [
            attribute id "Id": Integer [constraints (PrimaryKey)]
attribute context "Context": Integer [constraints(NotNull ForeignKey(e_Context))]
            attribute geometry "Geometry": GeoPoint [constraints (multiplicity "*")]  // LineString
            attribute fileName "Geometry File Name": FilePath  ]

        DataEntity e_ContextRefinement "Context Refinement": Parameter [
attribute id "Id": Integer [constraints (PrimaryKey)]
            attribute context "Context": Integer [constraints(NotNull ForeignKey(e_Context))]
            attribute geometry "Geometry": GeoPolygon
            attribute fileName "Geometry File Name": FilePath
            attribute cl "CL": Decimal ]
        DataEntity e_ContextAlignment "Context Alignment": Parameter [
            attribute id "Id": Integer [constraints (PrimaryKey)]
            attribute context "Context": Integer [constraints(NotNull ForeignKey(e_Context))]
            attribute geometry "Geometry": GeoPolyline [constraints (multiplicity "*")]  // LineString
attribute fileName "Geometry File Name": FilePath
            attribute cl "CL": Decimal ]

        DataEntity e_ContextBoundaryLine "Context Boundary Line": Parameter [
            attribute id "Id": Integer [constraints (PrimaryKey)]
attribute context "Context": Integer [constraints(NotNull ForeignKey(e_Context))]
            attribute geometry "Geometry": GeoPolyline //GeoPoint [constraints (multiplicity "*")] //FileField
            attribute fileName "Geometry File Name": FilePath
            attribute type "Type": DataEnumeration ContextBoundaryChoices [constraints (NotNull)]
            attribute dataType "Data type": DataEnumeration ContextBoundaryLineDataKindChoices [constraints (NotNull)] ]
        DataEntity e_ContextBoundaryPoint "Context Boundary Point": Parameter [
            attribute id "Id": Integer [constraints (PrimaryKey)]
            attribute contextBoundaryLine "Context Boundary Line": Integer [constraints(ForeignKey(e_Context))]
            attribute geometry "Geometry": GeoPoint
attribute fileName "Geometry File Name": FilePath   ]

        // Context's Sensors
        DataEntity e_ContextSensor "Context Sensor": Master [
            attribute id "Id": Integer [constraints (PrimaryKey)]
attribute boundaryPoint "Boundary Point": Integer [constraints(ForeignKey(e_ContextBoundaryPoint))]
            attribute sensor "Sensor": Integer [constraints(ForeignKey(e_Sensor))]
            attribute Description "Description": String [constraints (NotNull)]
            attribute associateDateTime "Associate Date Time": Datetime ]

// Context's Events
        DataEntity e_ContextEvent "Context Event": Document:Regular [
            attribute id "Id": Integer [constraints (PrimaryKey)]
            attribute context "Context": Integer [constraints(NotNull ForeignKey(e_Context))]
            attribute Name "Name": String [constraints (NotNull Unique)]
attribute type "Type": DataEnumeration ContextEventTypeChoices [constraints (NotNull)]
            attribute subType "Subtype": DataEnumeration ContextEventSubTypeChoices [constraints (NotNull)]
            attribute State "State": DataEnumeration ContextEventStateChoices [constraints (NotNull)]
```

```
        attribute startDateTime "Start Datetime": Datetime [constraints (NotNull)]
        attribute endDateTime "End Datetime": Datetime [constraints (NotNull)]
attribute Description "Description": String
        attribute returnPeriod "Return Period": Integer [defaultValue "1 "]
        attribute warmUp "Warm Up": Boolean [defaultValue "False"]
        attribute writingPeriodicity "Writing Periodicity": Decimal [defaultValue "1.0"]
        attribute writingPeriodicityUnit "Writing Periodicity Unit": String
attribute updateMaximumValue "Update Maximum Value": Decimal [defaultValue "1.0"]
        attribute updateMaximumValueUnit "Update Maximum Value Unit": String
        attribute hasSimulation "Has Simulation": Boolean [defaultValue "False"]
        attribute taskId "Task id": String
        attribute requester "Requester": Integer [constraints (ForeignKey(e_User))] ]
    DataEntity e_ContextEventResult "Context Event Result": Document:Weak [
        attribute id "Id": Integer [constraints (PrimaryKey)]
        attribute contextEvent "Context Event": Integer [constraints(ForeignKey(e_ContextEvent))]
        attribute maxDepth "Maximum Depth": GeoRaster
attribute maxLevel "Maximum Level": GeoRaster
        attribute maxQ "Maximum Q": GeoRaster
        attribute maxVel "Maximum Velocity": GeoRaster
        attribute time "Time": Datetime [constraints(NotNull)] ]
```

## Author contributions

All authors contributed to the article conception and design. The RiverCure Portal was designed by ARS, RF, JE and developed by AS, JM and IG. AMR performed material preparation and data collection. The manuscript was written by AMR, ARS and RF. All authors read and approved the final manuscript.

## Competing interests

The authors declare that they have no conflict of interest.

## Acknowledgements

This research is partially supported by Portuguese and European funds within the COMPETE 2020 and PORL-FEDER programs through project PTDC/CTA-OHR/29360/2017 RiverCure.

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
