# Peer review of "Flood simulation with the RiverCure approach: The open dataset of the Águeda 2016 flood event"

_Earth System Science Data, 2023_

## Referee Comment (RC1)

Review of the paper

Flood simulation with the RiverCure approach: The open dataset of the Águeda 2016 flood event

By Ana M. Ricardo et al.

The paper provides a use case for web-based high-level hydraulic analysis tool, designed to conduct numerical simulation from the data provided by user himself in order to access the outcomes of severe flooding. The case study suggests such assessment with the dataset on severe flooding event happened in February 2016 in the Agura River in Portugal.

The paper is a significant leap in shifting towards a new paradigm in hydrology (see e.g. (Rigon et al., 2022)): from Models as an Application (MaaA), e.g. installed on one's desktop, to Model as a Service (MaaS) – high-level environment designed for numerical experiments and results' visualization, located on the remote server and using graphical or API interface for user interaction (e.g. data handling and model setup).

The paper is well-organized and provides key features of the work motivation and the basic concept of the RiverCure portal, the data used for the numerical experiments and how the dataset is organized. Furthermore, apart from the input data the authors included the model output into the dataset as an instance of how the initial data could be utilized.

Having said that, I see several drawbacks in the dataset and the describing paper.

The paper structure could be improved. Section 2.2 Input data that describes only the spatial data is followed by Section 2.3 Output data, next Section 3 is called Data records and contains Section 3.1 Input data and Section 3.2 Output data again, which is very confusing. I suggest the authors combine the two sections to describe the data consistently – for input and output, spatial and temporal data separately.

The dataset handling should be improved. It took me a while to understand how I could download the data from the hydroshare.org website. The download process might be better documented for users not familiar with hydroshare.org or Bagit download tool. I suggest the authors prompt the download in Section 5 Data and code availability.

The dataset contents should also be improved. The listed spreadsheets *Agueda_hydrometric_PonteRedonda.xlsx* and *Agueda_hydrometric_Ribeiro.xlsx* contain only the streamflow discharge timeseries, contrary to what is stated on lines 159 – 167. The data spans for 16 days of hourly records at two gauges for streamflow discharge for the entire flooding event in February 2016

The river discharge and rain gauges locations could have been provided as a spatial coverage as well (e.g. geojson).

As of May 8[th], 2023, an attempt to load the layers to QGIS desktop 3.28.3 (Windows 10) via the provided links ended up as a failure (Web Map Service https://geoserver.hydroshare.org/geoserver/HS-937927473a3a4e66a07a2e2fdd9d581e/wms?request=GetCapabilities, Web Coverage Service https://geoserver.hydroshare.org/geoserver/HS-937927473a3a4e66a07a2e2fdd9d581e/wcs?request=GetCapabilities). Please check the data availability.

The http://rivercure.inesc-id.pt/ portal is well-designed but not very useful tool unless you get an instant guest access. Several days after I requested the access to DemoOrganization it is still pending. Without the access visiting the portal narrows to browsing some satellite maps. However, the RiverCure portal functionality may not be the main aim of the dataset and paper.

I suggest the authors address the mentioned issues, revise the dataset and the manuscript accordingly, and both could be accepted for publication after consistently improved.

Line-wise and figure-wise technical comments:

L157:    special – spatial

Fig. 4:    Please round the values in the map legend

References:

Rigon, R., Formetta, G., Bancheri, M., Tubini, N., D'amato, C., David, O., and Massari, C.: HESS Opinions: Participatory Digital eARth Twin Hydrology systems (DARTHs) for everyone - a blueprint for hydrologists, Hydrol. Earth Syst. Sci., 26, 4773–4800, https://doi.org/10.5194/hess-26-4773-2022, 2022.

---

## Author Comment (AC4)

*Dear Editor,*

*Flood simulation with the RiverCure approach: The open dataset of the Águeda 2016 flood event*

*By*

*Ana M. Ricardo, Rui M.L. Ferreira, Alberto Rodrigues da Silva, Jacinto Estima, Jorge Marques, Ivo Gamito, Alexandre Serra*

*The paper presents a WEB GIS platform "RiverCure", that is designed to manage data and to simulate floods using an in house developed code HiSTAV. The data set that is proposed for the Agueda 2016 flood event, it includes all the input data and the output data.*

*General Comments*

*Unfortunately, this paper possesses an in review paper in preprint (https://www.preprints.org/manuscript/202305.1472/v1), which includes a lot of information that is missing in the present manuscript, but it is not cited at least in review. In addition, more than half of the figures are very similar.*

**Authors:** We appreciate the valuable comments provided by the reviewer that we will fully address. The reviewer points to a paper that is effectively in review now. It has been in review since 19[th] May 2023, 60 days after the the submission of the present paper and its dataset. It is important to note that the paper mentioned focuses on a different aspect, complementing our work. In this publication, our primary objective is to present a unique and highly relevant dataset that can serve as a well-documented and reliable validation resource for modellers. On the other hand, the other paper primarily focuses on the key design aspects of the RiverCure Portal (RCP). Given the opportunity, we will further emphasize the significance of our proposed dataset while reducing the description of the RCP by referencing other relevant publications.

*Since the data are based on a poor quality DEM, I do not see the interest in sharing the data and the results, since a lot of issues must be linked to this modelling, some of the data is already in free access it seems. Furthermore, not enough information is provided to assess the quality of the data set and results especially when this argument is given: "The Águeda 2016 flood event was reported mainly on the Portuguese media allowing them to know reasonably well the limits of the flooded areas." The Web-GIS platform seems a very nice contribution, but unfortunately the dataset is not well described and poorly convincing. In addition, since the platform seems efficient, new data based on high-resolution DEM and better described hydraulic parameters, would be of larger interest and with new fast computer it is a nonsense to provide data that results of computation, since models are evolving.*

**Authors:** We accept that we did not elaborate enough on the significance of the proposed dataset. We are confident that this dataset is indeed relevant, and we are willing to improve the manuscript in this key aspect.

It is true that the flooded area has been documented in the media and is well-known, at least in Águeda downtown. What is not known is why Águeda has been flooded, because the river level never exceeded the protection walls.

To be clear, modellers using a DTM other than this would not be able to reproduce the flood, unless they tweak resistance coefficients to wrong values.

This manuscript addresses and solves this issue – the vulnerability in the river walls was detected through field work and included in this DTM.

The results of the hydrodynamic model are not included to showcase its performance. They are needed to reconstruct velocities and, knowing the channel geometry, discharges at Águeda bridge and along Águeda waterfront.

We emphasise that flooding is a recurrent problem of Águeda city. There are no curated dataset that allows for the correct description of its flooding events. The dataset we are proposing includes hydrometric, udometric and numerically-produced data from a calibrated model and thus constitutes a relevant and complete validation test for other flood models and a tool to better mitigate floods in this river and in similar rivers.

Per se, none of the data sets (hydrometric, udometric or numerically-produced) would be sufficient. It is the synthesis that matters and explaining it is the driver of this paper.

We insist that the numerically-produced data is a closure – we know the maximum flow depth (from the media and official communications) and the boundary conditions but we do not know the discharge at Águeda bridge because the hydrometric station was malfunctioning during the event. The discharge at Águeda bridge had to be estimated with the hydrodynamic model. This task required a time-consuming assemblage of topo-bathymetric data and calibration procedure. Since the city was flooded not because of the protection walls overtopping but because of vulnerabilities in these walls, these had to be investigated by us to be incorporated in the DEM. Thus, the DEM was based on the global ASTER-DEM but a relatively demanding work was carried out to obtain the final DEM included in the dataset to ensure an adequate resolution for the river bathymetry and local vulnerabilities. This involved collecting and updating data regarding the elevation of the protective walls of the riverfront of Águeda city and reproducing the geometry of the under-passages of Águeda bridge. Extra details will be updated on the dataset with the revised version of the manuscript.

*Specific comments*

*Line 26: here systemic vulnerability will be a better word.*

**Authors:** We will change accordingly. Thank you for the suggestion.

*Figure 2 does not include enough details since it is linked to the data.*

**Authors:** Figure 2 refers to RCP structure therefore we may remove it in the revised version and refer to the dedicated publication.

*Line 92: please give the full definition name of CFL I guess: Courant–Friedrichs–Lewy*

**Authors:** Thank you for spotting the no identified abbreviation. Indeed, CFL stands for Courant-Friedrichs-Lewy, a conventional condition to restrict the computational time step. We will include the full name on the revised version.

*Figure 4: improve please.*

**Authors:** We will improve the image quality, the colour scale and the legend.

*Line 197: sampled hydrograph is confusing when used for modelling, what about ground truth?*

**Authors:** By sampled hydrograph we meant the hydrograph obtained by modelled data. We will improve the writing style to make it clear. There was a gauge station within the modelled domain, at Águeda Bridge, however it has been malfunctioning for few year, including during the flood event. Therefore, there is no measured data available for comparing hydrographs. In our opinion, this particularity strengthens the relevance of the presented dataset, that includes a synthesis of hydrometric, udometric and numerically-produced data from a calibrated model.

---

## Author Response (AR1)

**Reviewer #1**

*The paper provides a use case for web-based high-level hydraulic analysis tool, designed to conduct numerical simulation from the data provided by user himself in order to access the outcomes of severe flooding. The case study suggests such assessment with the dataset on severe flooding event happened in February 2016 in the Agura River in Portugal.*

*The paper is a significant leap in shifting towards a new paradigm in hydrology (see e.g. (Rigon et al., 2022)): from Models as an Application (MaaA), e.g. installed on one's desktop, to Model as a Service (MaaS) – high-level environment designed for numerical experiments and results' visualization, located on the remote server and using graphical or API interface for user interaction (e.g. data handling and model setup).*

*The paper is well-organized and provides key features of the work motivation and the basic concept of the RiverCure portal, the data used for the numerical experiments and how the dataset is organized. Furthermore, apart from the input data the authors included the model output into the dataset as an instance of how the initial data could be utilized.*

**Authors:** We thank the reviewer for the positive assessment of our article. We believe the revised version is significantly improved, accommodating the valuable suggestions. We also thank the Rigon et al., (2022) reference suggestion; indeed, our efforts/concerns related to RiverCure Portal are fully aligned with the DARTHs paradigm.

*Having said that, I see several drawbacks in the dataset and the describing paper.*

*The paper structure could be improved. Section 2.2 Input data that describes only the spatial data is followed by Section 2.3 Output data, next Section 3 is called Data records and contains Section 3.1 Input data and Section 3.2 Output data again, which is very confusing. I suggest the authors combine the two sections to describe the data consistently – for input and output, spatial and temporal data separately.*

**Authors:** We thank the reviewer for the suggestion. We agree with the suggestions of the reviewer. We changed the structure to grant a straightforward understanding of the dataset and the software tools employed. That includes merging subsections 2.2 and 3.1 and the subsections 2.3 and 3.2. The details regarding the software tools are provided in a dedicated section. Also the dataset was re-organized by nature of the data. Those improvements resulted in the following structure:
    1. Introduction
    2. Supported software tools
    3. Data records
        3.1 Overview of the dataset
        3.2. Geometrical features
        3.3 Field data
            3.3.1. Hydrological data
            3.3.2 Topo-bathymetric data
            3.3.3. Land-use data
        3.4 Hydrometric data
            3.4.1 Envelope of maxima
            3.4.2 Time-lapse
    4 Data validation
    5 Data and code availability
    6 Conclusions

*The dataset handling should be improved. It took me a while to understand how I could download the data from the hydroshare.org website. The download process might be better documented for users not familiar with hydroshare.org or Bagit download tool. I suggest the authors prompt the download in Section 5 Data and code availability.*

**Authors:** We thank the author for the suggestion. We added further details the download process in Section 5. We also included a figure (Figure 2 in the manuscript) with a schematic representation of the dataset organization including the file names.

*The dataset contents should also be improved. The listed spreadsheets Agueda_hydrometric_PonteRedonda.xlsx and Agueda_hydrometric_Ribeiro.xlsx contain only the streamflow discharge timeseries, contrary to what is stated on lines 159 – 167. The data spans for 16 days of hourly records at two gauges for streamflow discharge for the entire flooding event in February 2016*

**Authors:** We improved the dataset description regarding the hydrological data. The data measured in the two gauges was water levels that was converted into discharge, which is the input data for the numerical model HiSTAV. The revised version of the manuscript includes further details on the discharge computation (section 3.3.1).

*The river discharge and rain gauges locations could have been provided as a spatial coverage as well (e.g. geojson).*

**Authors:** Thank you for the suggestion. We added those locations as recommended.

*As of May 8th, 2023, an attempt to load the layers to QGIS desktop 3.28.3 (Windows 10) via the provided links ended up as a failure (Web Map Service https://geoserver.hydroshare.org/geoserver/HS-937927473a3a4e66a07a2e2fdd9d581e/wms?request=GetCapabilities, Web Coverage Service https://geoserver.hydroshare.org/geoserver/HS-937927473a3a4e66a07a2e2fdd9d581e/wcs?request=GetCapabilities). Please check the data availability.*

**Authors**: We are sorry about the experienced difficulty accessing the data. We have carefully checked the data availability on Hydroshare. The download ran in the background without any progress feedback and took several minutes, but no failure was detected.

*The http://rivercure.inesc-id.pt/ portal is a well-designed but not very useful tool unless you get an instant guest access. Several days after I requested the access to DemoOrganization it is still pending. Without the access visiting the portal narrows to browsing some satellite maps. However, the RiverCure portal functionality may not be the main aim of the dataset and paper.*

**Authors**: Over the last months, we have been working on improving the efficiency of the RiverCure Portal, including a host server transfer. The portal was temporarily unavailable. and some changes have been introduced. That was the main reason for the pending request. To avoid any issue related to confidentiality and for the reviewers' ease, we propose to provide login details to the reviewers by sending them to the Editor. Although the RCP functionality is not the main aim of this paper, we believe it has the potential to become an important tool for hydrologic and hydraulic applications; therefore, we want to ensure the reviewers can access it.

*I suggest the authors address the mentioned issues, revise the dataset and the manuscript accordingly, and both could be accepted for publication after consistently improved.*

*Line-wise and figure-wise technical comments:*

*L157:    special – spatial*

*Fig. 4:    Please round the values in the map legend*

**Authors**: The typo and the map legend are fixed. We thank, again, the positive feedback and all the suggestions provided by the reviewer that certainly contributed to an improved version of our paper.

*References:*
*Rigon, R., Formetta, G., Bancheri, M., Tubini, N., D'amato, C., David, O., and Massari, C.: HESS Opinions: Participatory Digital eARth Twin Hydrology systems (DARTHs) for everyone - a blueprint for hydrologists, Hydrol. Earth Syst. Sci., 26, 4773–4800, https://doi.org/10.5194/hess-26-4773-2022, 2022.*

**Reviewer #2**

*In their manuscript "Flood simulation with the RiverCure approach: The open dataset of the Águeda 2016 flood event", Ricardo and colleagues present a web platform designed to integrate hydrodynamic and morphodynamic modelling tools and input data, together with an exemplary application in the context of a hydrodynamic simulation of the Águeda river flood in 2016. For this purpose, the authors provide both the input data (DEM, surface roughness, model domain) and the model output (water level, flow velocity, etc.).*

*The manuscript is fairly well written and organised. I appreciate the effort of the River Cure Portal (RCP) as a platform to allow for the web-based integration of data and models in the context of hydrodynamic simulations. The data is accessible via https://www.hydroshare.org which I was not aware of before, but which appears to adhere to FAIR principles. The dataset itself, though, is insufficiently documented in some parts, and the presentation of the data in the manuscript appears partly incorrect (see below under technical comments).*

**Authors:** We are thankful for the reviewer comments that we fully addressed. Those have, certainly, contributed to the improvement of our paper. We are carrying on efforts to improve the dataset documentation. In the revised version of the manuscript, a clarification on the hydrological data is added as well as the description of the conversion into discharge data (section 3.3.1.)

*The main issue of this contribution is, however, that the presented dataset does not at all meet the review criterion of "significance", neither with respect to the aspects of "uniqueness" nor "usefulness". As for "uniqueness", the model input data for the case of the Águeda river flood largely consists of data which is available to the public via other channels (the global ASTER-DEM, public land use and land cover data from which Manning-Strickler's friction coefficient is derived, public rain gauge (or discharge?) records). The usage and pre-processing of these data to produce the case study input is undemanding. The output data, in turn, cannot be considered unique either, as it can be re-produced by any hydrodynamic model at moderate computational cost. In fact, the existence of RCP makes it even easier to produce the published model output. The criterion of "usefulness" is of course more subjective, but I do not see that the published dataset allows for any new approaches to address pressing research questions in the field. The authors themselves do not highlight any specific research questions or application cases for the data except that it "[..] can be used as a starting point to design other experiments and tools, and to explore the RiverCure Portal".*

**Authors:** We thank the reviewer's comments. We did our best to further improve the manuscript. We are confident that the presented dataset is relevant from both uniqueness and usefulness point of view. At the same time, we accept that we did not elaborate enough on the significance of the proposed dataset on the first version of the manuscript. The revised version of the manuscript clarifies our objective and the relevance of the data.

Flooding is a recurrent problem of Águeda city. However, there are no curated dataset that allows for the description of its flooding events. The 2016 flood event is of particular interest due to the large media coverage allowing to gather informal but accurate data regarding water depth and extent of flooded areas – in this event, the water depth in the river, near the bridge was, at its peak, from 10 to 20 cm below the protection wall crest. This has been observed by local authorities and registered in photos and videos between 4 and 8 PM on the 12th February.

The city was flooded not because of overtopping the protection walls but because of vulnerabilities in these walls. These vulnerabilities have been investigated by us and incorporated in the DEM. We do not know of other dataset that incorporates these features – in that sense this is unique. It should be noticed, at this point, that the hydrometric station placed at River Águeda bridge was malfunctioning during the February 2016 event. Thus, this is the only set of data that describes this event and it requires the numerically produced data, from the calibrated hydrodynamic tool, to complete the flood description, providing discharges and flow velocities. This effort synthetizes topo-bathymetric and numerically produced data and is very relevant locally as we are documenting an event relevant for River Águeda stakeholders.

But it is also relevant for a wider community of flood modellers. Since we have time series of discharges in upstream reaches and a rather accurate estimate of the peak water elevation in Águeda bridge, this data set configures a well-documented validation event for flood models, particularly, shallow-water models. This is our main aim – to provide the flood modelling with a well-documented flood event to validate flood models meant to operate in similar conditions.

One last note regarding the effort involved in producing the data. The DEM was based on the global ASTER-DEM. Still, a relatively demanding work was carried out to obtain the final DEM included in the dataset to ensure an adequate resolution for the river bathymetry. We conducted field-work to detect non-evident vulnerabilities, thus avoiding errors that other topographic models might have. This involved collecting and updating data regarding the elevation of the protective walls of the riverfront of Águeda city and reproducing the geometry of the under-passages of Águeda bridge. Extra details were updated on the dataset with the revised version of the manuscript.

*I would like to emphasise, again, that I welcome the concept and implementation of the RCP (although I cannot say how it compares to alternatives frameworks). However, neither the design of the RCP nor the included HiSTAV model are within the scope of ESSD which is about the publication of unique and useful data. I therefore recommend rejecting the manuscript.*

**Authors:** We are not attempting to publish the RCP design or HiSTAV features in this paper. We have been doing that in other papers. We intent to publish a unique and relevant dataset, to be used by modellers as a well-documented and trusted validation event. Per the Journal's rules, we must to include details on all software tools to ensure that numerical data is reproducible. Attempting to address the reviewer's concerns, the revised manuscript further highlights the features of the dataset and corrects any unreasonable descriptions of software tools, including RCP and HiSTAV.

*Apart from that, I have a few technical comments which are, however, not to be considered comprehensive:*

- *The data in the data repository is insufficiently documented. Not all files contain metadata, and there is no document that provides an overarching overview of the dataset as an entrypoint to users.*

**Authors:** We are working on the documentation of the dataset. Following the reviewer suggestion, an overview of the dataset is provided.

- *Section 3.1 (3): this subset of data is desribed as precipitation data (the term "udometric" is highly uncommon), but the actual data files appear to contain discharge data - quite confusing. None of these files contain any metadata.*

**Authors:** The data measured in the two gauges was water levels that was converted into discharge. For clarity, the revised version of the manuscript includes further details on this type of input data (section 3.3.1). We replaced the term 'udometric'.

- *Appendix: The RSL-based "Context"-definition of the data might be interesting within the RCP, but I think it is irrelevant in the context of the data presentation.*

**Authors:** We understand the reviewer observation. Nevertheless, we consider that a rigorous specification of that data model could be relevant to some readers, who may be interested in further analyse such details.

*l. 68: "included in the above input data list" - where is that?*

**Authors:** The sentence is corrected.

- *l. 71: "altimetry" - do you mean DEM?*

**Authors:** Yes, we meant DEM. The sentence is improved.

- *l. 136: "periodicity"? Do you mean "interval"?*

**Authors:** We meant to state that the RCP users can define the simulated time interval between results writing. The sentence is changed to improve readability (lines 201-202)

- *l. 208: the reference Ricardo et al. (2022) is missing in the reference list*

**Authors:** This reference refers to the dataset published in Hydroshare. The reference list will be carefully checked in the revised manuscript.

**Reviewer #3**

*The manuscript describes an approach to combine data analysis and simulation as a service. To offer simulations as services is in principle a new way forward and may pave the way for many applications in natural systems analysis.*

*The manuscript shows a case study of a re-simulated flood event in 2016 in the Agueda river basin. In my opinion, the innovative aspect of the presented case study lays in the combination of data and simulation services, rather than in the dataset itself.*

*The dataset itself offers an opportunity for studying the flood impacts, e.g. by overlaying flood depths with the damaged buildings to elaborate vulnerability functions or to study the disruption of human mobility during the flood event. Thus, the dataset is worth to describe and being published.*

*However, the manuscript and also the data need to be improved before being useable by other scientists or practitioners. In the following, I am listing the main critical points that should be considered on the revision of the manuscript:*

**Authors:** We thank the reviewer for the positive assessment of our article. We placed our best efforts to further improve the work accommodating the valuable suggestions.

*- A more stringent division/distinction between data and modeling is needed. At the moment, the method (the model) is in the foreground. A restructuring of the manuscript in that the data itself becomes a more prominent position might be worth to consider.*

**Authors:** We changed the manuscript' structure to separate the descriptions of the dataset and the software tools employed. Moreover, we elaborated on the significance of the proposed dataset. We note that the dataset we are proposing is a synthesis of topo-bathymetric, hydrological, and numerically-produced hydrometric data. The latter is a closure – we know the maximum flow depth and the boundary conditions but we do not know the discharge at Águeda bridge because the hydrometric station was malfunctioning during the event. The discharge at Águeda bridge had to be estimated with the hydrodynamic model. This task that requires a time-consuming assemblage of topo-bathymetric data and calibration procedure. The relevance of this numerically-produced data may have been overemphasized due to the relatively higher difficulty in producing it. We corrected this aspect in the final version of the manuscript.

*-The validation of the output data must be much more improved. At least, the simulated flow depths must be compared with observed ones at sample sites. If there is a dataset of affected buildings or blocked roads available, the output dataset can be compared with these proxy data that can be used for validation.*

**Authors:** This is a crucial issue indeed. We have testimonials of local civil protection agents and photos that show, with no uncertainty, that the peak water level, registered on the 12th February between 16:00 and 20:00, never overtopped Águeda's protection walls. The level stayed within 10 to 20 cm below the crest of the walls at Águeda bridge. This is the only water level estimate we have since the hydrometric station was malfunctioning. Since we have the bathymetry and since we have reliable hydrometric data at upstream boundaries, we used this information to calibrate the model and compute velocities and discharges at Águeda bridge. The dataset comprises boundary information, topo-bathymetric information, calibrated roughness coefficients and computed discharges and velocities. So, to be clear, and answering the reviewer, the simulated flow levels have been compared with the observed ones at the only sample site available – downtown Águeda. But not to validate the model – such comparison is what allows for model calibration and is thus the condition of possibility of having the dataset complemented with numerically-produced discharges at Águeda bridge. In our opinion this is a strong point of the manuscript – we provide a synthesis of topo-bathymetric, hydrological, and numerically-produced hydrometric data from a calibrated model, that constitute a relevant and complete validation test for other flood models and a tool to better mitigate floods in this river and in similar rivers. Per se, none of the data sets (topo-bathymetric, hydrological or numerically-produced) would be sufficient. It is the synthesis that matters and explaining it is the driver of this paper.

Following the reviewer' suggestion, we improved the text so that our arguments come across more clearly.

*- It would be of interest if the data portal can also store sensor data that document the flood event (eg., event documentation data, drone images, flow depth measurements, sediment transport and deposition, changes in cros-section profiles, etc.). This would make the dataset more interesting and valuable for developing and evaluating (benchmarking) newly developed simulation models.*

**Authors:** The RCP is currently not storing that type of data. However, we are working along these lines to include data from "social sensors" (dedicated channels to collect images from social networks and its metadata). The aim is to transform such data into information that can be assimilated into hydrodynamic tools. This, however, is not the main focus of this paper – this is meant to describe the Feb 2016 Águeda flood, synthetizing all available data, including informal data. The planned advances should be published in a different type of publication.

*- The main criticism of the simulation model is the poor representation of the river channel geometry. It is not clear which cross-section data was used. The simplification of the overall assumed channel depth of 5 m is not accurate. The presentation of local case study data is valuable only if it is really accurate and at a high spatial resolution. This approach of river geometry simplification is used for global models but not for local-scale models - or for river reaches that are dominated by downstream sea or lake levels. River conveyance capacity is one of the most sensitive factors in flood risk analysis. Moreover, please add more information on the mesh size. This all would make the data more comprehensible and more useful for others.*

**Authors:** We agree with the reviewer on the importance of simulating adequately river conveyance. But we disagree that we have a poor representation of the river channel geometry. River Águeda in front of Águeda town has suffered many interventions that resulted in a rather simple geometry – River Águeda is indeed a channel with a rectangular cross-section with protection walls. In the revised manuscript, we provided a more detailed description of the process for defining the river bathymetry obtained by elements provided by Águeda municipality. These elements confirm the accuracy and the adequacy of the spatial resolution of the DEM in what concerns the river bottom.

*Minor points:*

*-The abbreviation HiSTAV must be explained before being used the first time and the model must be cited.*

**Authors:** HiSTAV stands for the High Performance Computing version of Strong Transients in Alluvial Channels. The model was originally designed to tackle river morphology problems, especially when involving subcritical-supercritical shocks or transitions, as is the case in anthropic floods such as those resulting from a dam-break. The morphological module is not explored in this work but we see no reason to change the name of the model.

*-Figure 1: Add a few explanations in the figure captions, explain abbreviations.*

**Authors:** Figure 1 was removed as the details on the RCP have been published recently.

*-line 86: what is an "in-house" model*

**Authors:** It is a model developed by the authors of the study, as opposed to a model (freeware or proprietary) developed elsewhere. In this case, the efforts of developing HiSTAV have been coordinated by co-author Rui M.L. Ferreira. Details of its HPC implementation can be consulted in Conde et al. (2020).

*-line 138: Please explain why the correction factor +0.5 has been chosen*

**Authors:** The physical parameter used to define human stability and manoeuvrability thresholds is the mass flux per unit width and per unit density in the direction of the flow. We named as hazard index the variable defined as $h×(v+0.5)$ included in HiSTAV, where $h$ is the flow depth, and $v$ is the flow velocity, following research by colleagues in the UK and TU Delft (see for example Penning-Rowsell et al, 2015 or Jonkman and Penning-Rowsell, 2008).

*-line 179: "bam break"?*

**Authors:** It was a typo. We meant 'dam-break'.

*-line 182: Who used the data from the media to map the limits of the flooded areas? Has this been done for validation?*

**Authors:** The extensive media coverage of the 2016 flood event in Águeda allowed a significant collection of images with vertical references. Data from the media was used primarily to ascertain the water level at Águeda bridge, once the hydrometric station was malfunctioning. It was also used to verify the extent of inundation along Rua Luís de Camões and Rua Vasco da Gama, as there are photos taken between 16:00 and 20:00 on the 12th February. The simulated flow levels have been compared with the observed ones at the sample site available but have not been used to validate the model. That comparison allows for model calibration and is thus the condition of the possibility of having the dataset complemented with numerically-produced discharges at Águeda bridge.

*-figure 3: add scale bar and a better legend. Please add also the site of the gauging station.*

**Authors:** This figure has been removed but its contents were included in an improved version in Figure 1 of the revised manuscript. The location of the gauging station is also added as a georeferenced layer into the dataset.

*-figure 4: add scale bar and a proper legend*

**Authors:** The figure has been improved (Figure 5 in the revised version).

*-How did you asess the role of insflow 1 and 2 from the measured flow in the gauging station?*

**Authors:** The discharge time-series at the two gauges were input boundary conditions of the model.

-*Figure 5: Please add the toponyms "Ponte Redonda" and "Ribeiro" in figure 3*

**Authors:** The location of Ponte Redonda and Ribeiro stations were included in Figure 1.

*- Please describe the REST API in more detail. This would be an innovative feature for data sharing and especially for interoperability.*

**Authors:** The article's main goal is to present a well-documented, unique and relevant dataset, that is publicly available through HydroShare, a worldwide repository. The REST API is a feature of the repository, therefore, its detailed explanation will be out of the scope of the article. Nevertheless, we included some extra information about Hydroshare. For further details regarding the API are provided at https://help.hydroshare.org/introduction-to-hydroshare/getting-started/use-the-api/.

References:

Conde, D., Canelas, R.B., and Ferreira, R.M.L. (2020). A unified object-oriented framework for CPU+GPU explicit hyperbolic solvers. *Advances in Engineering Software* 148, 102802. DOI: j.advengsoft.2020.102802

Penning-Rowsell, E.C., P. Floyd, D. Ramsbottom, and S. Surendran (2005). Estimating Injury and Loss of Life in Floods: A Deterministic Framework. *Natural Hazards* 36:43-64. DOI: 10.1007/s11069-004-4538-7

Jonkman, S. N. and Penning-Rowsell, E. (2008). Human Instability in Flood Flows. *JAWRA Journal of the American Water Resources Association* 44(5), 1208–1218. DOI:10.1111/j.1752-1688.2008.00217.x

**Reviewer #4**

*Dear Editor,*

*Flood simulation with the RiverCure approach: The open dataset of the Águeda 2016 flood event*

*By*

*Ana M. Ricardo, Rui M.L. Ferreira, Alberto Rodrigues da Silva, Jacinto Estima, Jorge Marques, Ivo Gamito, Alexandre Serra*

*The paper presents a WEB GIS platform "RiverCure", that is designed to manage data and to simulate floods using an in house developed code HiSTAV. The data set that is proposed for the Agueda 2016 flood event, it includes all the input data and the output data.*

*General Comments*

*Unfortunately, this paper possesses an in review paper in preprint (https://www.preprints.org/manuscript/202305.1472/v1), which includes a lot of information that is missing in the present manuscript, but it is not cited at least in review. In addition, more than half of the figures are very similar.*

**Authors:** We are thankful for the reviewer comments that we have fully addressed. The mentioned in review paper was submitted after the submission of the present paper and its dataset. The two publications are complementary but with different and well defined focus. Here we intend to publish a unique and relevant dataset, to be used by modellers as a well-documented and trusted validation event. The other paper presents the key design aspects of the RiverCure Portal (RCP). Here, we have further elaborated on the significance of the proposed dataset and reduced the description of the RCP citing other publications.

*Since the data are based on a poor quality DEM, I do not see the interest in sharing the data and the results, since a lot of issues must be linked to this modelling, some of the data is already in free access it seems. Furthermore, not enough information is provided to assess the quality of the data set and results especially when this argument is given: "The Águeda 2016 flood event was reported mainly on the Portuguese media allowing them to know reasonably well the limits of the flooded areas." The Web-GIS platform seems a very nice contribution, but unfortunately the dataset is not well described and poorly convincing. In addition, since the platform seems efficient, new data based on high-resolution DEM and better described hydraulic parameters, would be of larger interest and with new fast computer it is a nonsense to provide data that results of computation, since models are evolving.*

**Authors:** We accept that we did not elaborate enough on the significance of the proposed dataset. We are confident on the data relevance and have improved the manuscript in this key aspect.

Flooding is a recurrent problem of Águeda city. However, there are no curated dataset that allows for the description of its flooding events. The dataset we are proposing includes topo-bathymetric, hydrological or numerically-produced hydrometric data from a calibrated model and constitutes, in our opinion, a relevant and complete validation test for other flood models and a tool to better mitigate floods in this river and in similar rivers. Per se, none of the data sets (topo-bathymetric, hydrological or numerically-produced) would be sufficient. It is the synthesis that matters and explaining it is the driver of this paper.

The numerically-produced data is a closure – we know the maximum flow depth and the boundary conditions but we do not know the discharge at Águeda bridge because the hydrometric station was malfunctioning during the event. The discharge at Águeda bridge had to be estimated with the hydrodynamic model. This task required a time-consuming assemblage of topo-bathymetric data and calibration procedure. Moreover, in the 2016 flood event, the city was flooded not because of the protection walls overtopping but because of vulnerabilities in these walls. These vulnerabilities have been investigated by us and incorporated in the DEM. Thus, the DEM was based on the global ASTER-DEM but a relatively demanding work was carried out to obtain the final DEM included in the dataset to ensure an adequate resolution for the river bathymetry and local vulnerabilities. This involved collecting and updating data regarding the elevation of the protective walls of the riverfront of Águeda city and reproducing the geometry of the under-passages of Águeda bridge. Extra details were updated on the dataset with the revised version of the manuscript.

*Specific comments*

*Line 26: here systemic vulnerability will be a better word.*

**Authors:** We changed accordingly. Thank you for the suggestion.

*Figure 2 does not include enough details since it is linked to the data.*

**Authors:** The figure referred to RCP structure therefore we removed it in the revised version and referred to the dedicated publication.

*Line 92: please give the full definition name of CFL I guess: Courant–Friedrichs–Lewy*

**Authors:** Thank you for spotting the no identified abbreviation. Indeed, CFL stands for Courant-Friedrichs-Lewy, a conventional condition to restrict the computational time step. We included the full name on the revised version.

*Figure 4: improve please.*

**Authors:** We improved the image quality and the legend (it is now Figure 5).

*Line 197: sampled hydrograph is confusing when used for modelling, what about ground truth?*

**Authors:** By sampled hydrograph we meant the hydrograph obtained by modelled data. We improved the writing style to make it clear. There was a gauge station within the modelled domain, at Águeda Bridge, however it has been malfunctioning for few years, including during the flood event. Therefore, there is no measured data available for comparing hydrographs. In our opinion, this particularity strengthens the relevance of the presented dataset, that includes a synthesis of topo-bathymetric, hydrological or numerically-produced hydrometric data from a calibrated model.

---

## Editor Decision (ED1)

Dear Authors:

I have received five reviews of your manuscript. The first Referee has not provided any substantial criticism. The second Referee has provided extensive comments and recommended re-considering the manuscript after major revision. The 3rd and 4th reviewers recommended that the manuscript be rejected. Criticisms of all Referees were addressed in your responses. I carefully read their arguments, as well as your answers, and came to the conclusion that your position is well founded and your responses successfully smooth all the main criticisms of the Referees. I appreciate your clear responses. As a result, I asked the second Referee to provide additional review of the revised manuscript and received the review. I am pleased to note that the manuscript has been significantly improved, and only technical changes are recommended at this stage:

the figures containing maps (fig 1 and fig 5) need to be improved. Please add a scale bar or a coordinate grid at the outside borders of the map for orientation.

After these minor edits, I recommend the manuscript for publication.

Alexander Gelfan,
ESSD Editor

---

## Author Response (AR2)

**Reply to Editor and to reviewer**

We are thankful to the Editor for the opportunity to improve our manuscript. We also thank the assessment of the Reviewer 2 and the Editor. As suggested, we have improved the figures with maps by adding a scale bar and by adding the labels directly on the SIG tool.

We thank again all the reviewers for their comments and criticism, it allow us so significantly improve the manuscript and make our point clearer.